# Incidence and Risk Factors for Blood Stream Infection in Mechanically Ventilated COVID-19 Patients

**DOI:** 10.3390/antibiotics11081053

**Published:** 2022-08-03

**Authors:** Konstantinos Mantzarlis, Konstantina Deskata, Dimitra Papaspyrou, Vassiliki Leontopoulou, Vassiliki Tsolaki, Epaminondas Zakynthinos, Demosthenes Makris

**Affiliations:** Department of Critical Care, University Hospital of Larissa, School of Medicine, University of Thessaly, 41110 Thessaly, Greece; kostadv@gmail.com (K.D.); dimitra.papaspyrou@hotmail.com (D.P.); vasoula_leontop@yahoo.com (V.L.); vasotsolaki@yahoo.com (V.T.); ezakynth@yahoo.com (E.Z.); appollon7@hotmail.com (D.M.)

**Keywords:** SARS-CoV-2 infection, mechanical ventilation, risk factors, blood stream infection, mortality

## Abstract

It is widely known that blood stream infections (BSIs) in critically ill patients may affect mortality, length of stay, or the duration of mechanical ventilation. There is scarce data regarding blood stream infections in mechanically ventilated COVID-19 patients. Preliminary studies report that the number of secondary infections in COVID-9 patients may be higher. This retrospective analysis was conducted to determine the incidence of BSI. Furthermore, risk factors, mortality, and other outcomes were analyzed. The setting was an Intensive Care Unit (ICU) at a University Hospital. Patients suffering from SARS-CoV-2 infection and requiring mechanical ventilation (MV) for >48 h were eligible. The characteristics of patients who presented BSI were compared with those of patients who did not present BSI. Eighty-four patients were included. The incidence of BSI was 57%. In most cases, multidrug-resistant pathogens were isolated. Dyslipidemia was more frequent in the BSI group (*p <* 0.05). Moreover, BSI-group patients had a longer ICU stay and a longer duration of both mechanical ventilation and sedation (*p* < 0.05). Deaths were not statistically different between the two groups (73% for BSI and 56% for the non-BSI group, *p* > 0.05). Compared with non-survivors, survivors had lower baseline APACHE II and SOFA scores, lower D-dimers levels, a higher baseline compliance of the respiratory system, and less frequent heart failure. They received anakinra less frequently and appropriate therapy more often (*p <* 0.05). The independent risk factor for mortality was the APACHE II score [1.232 (1.017 to 1.493), *p* = 0.033].

## 1. Introduction

Severe acute respiratory syndrome coronavirus 2 (SARS-CoV-2) was first identified in Wuhan, China, in December 2019 [1]. Intensive care unit (ICU) admission is required for 20% of patients with coronavirus disease 2019 (COVID-19) due to acute respiratory distress syndrome (ARDS) or other complications [2,3,4].

The incidence of blood stream infections (BSIs) among non-COVID-19 patients with infection is high [5]. The immune dysregulation induced by severe SARS-CoV-2 infection and the immunosuppressive agents used for treatment can predispose patients to concurrent infections. Studies detected a reduction in both CD4+ T and CD8+ T lymphocyte counts, an increase in neutrophils, a reduction in interferon gamma (IFN-γ) serum concentrations, and a cytokine pattern characterized by excess pro-inflammatory molecules [6,7,8]. Moreover, the need for vasopressors, renal replacement therapy (RRT), or sometimes extracorporeal membrane oxygenation (ECMO) may increase the risk of developing infectious complications.

There are reports that the incidence of BSIs is higher for COVID-19 patients in comparison with non-COVID-19 patients [9] during the ICU stay. However, there is scarce data regarding secondary infections in patients with severe COVID-19 [10,11,12], especially for those admitted to the ICU who receive invasive mechanical ventilation. There is also limited evidence on how secondary infections and especially BSIs affect patients’ outcomes, such as mortality, duration of mechanical ventilation, or length of stay.

We therefore aimed to assess the incidence rate, identify risk factors for the first episode of BSI, and determine survival and other outcomes in COVID-19, mechanically ventilated patients.

## 2. Materials and Methods

This is a retrospective analysis of prospectively collected data. The study was conducted from 1 April to 31 December 2020 in an ICU dedicated to patients suffering from SARS-CoV-2 infection and requiring invasive mechanical ventilation in the University Hospital of Larissa, Thessaly, Greece. Inclusion criteria were: (a) ICU admission for SARS-CoV-2 infection, and (b) intubation and mechanical ventilation for >48 h. Exclusion criteria were: (a) age <18 years old, and (b) ICU readmission. The first episode of BSI was reported. Patients were divided in two different groups: the first group consisted of patients that presented BSI, and the second one of patients without BSI.

### 2.1. Outcome

The primary outcome of this study was the incidence of ICU-acquired BSIs in COVID-19, mechanically ventilated, critically ill patients. The secondary outcome was the identification of risk factors for the first episode of BSI.

### 2.2. Clinical Assessment

For all study patients, the following characteristics were recorded: age, sex, characteristics of the respiratory system, illness severity based on Acute Physiology and Chronic Health Evaluation Score II (APACHE II), Sequential Organ Failure Assessment (SOFA) score at admission, history of hospitalization during the last 3 months before current admission, history of invasive procedures (gastroscopy, colonoscopy, bronchoscopy, or surgery), medical history, history of antibiotic use, type and duration of antibiotics used, and finally therapies and laboratory findings related to COVID-19 infection. For survivors and non-survivors, several characteristics that might affect mortality were also taken into account.

### 2.3. Microbiology

Identification and susceptibility testing of the isolated pathogens were performed by the Vitek 2 automated system (bioMerieux, Marcy l’ Etoile, France). For the interpretation of the results, EUCAST breakpoints were used.

### 2.4. Statistical Analysis

The results are presented as the frequency (%) for categorical variables or the median (25th, 75th quartiles) for continuous variables. The normality of data distribution was assessed by a Kolmogorov/Smirnov test. Categorical variables were compared using a chi square test or Fisher’s exact test where appropriate; continuous variables were compared by a Mann–Whitney *U* test. Multivariate analyses were performed to determine variables associated with BSI or mortality. Only variables with a *p* value <0.05 were used in the stepwise logistic regression models. The analysis was performed between two groups (patients with BSI and patients without BSI). Exposure to potential risk factors was taken into account only before diagnosis of infection. A mortality analysis was performed between two groups (survivors and non survivors). SPSS software (SPSS 17.0, Chicago, IL, USA) was used for the data analysis.

## 3. Results

A total of 90 cases were studied. One case was a readmission, and data for five cases were incomplete, leaving 84 cases for the analysis. The characteristics of participants are presented in Table 1, Table 2 and Table 3. The incidence of BSIs was 57%, since 48 patients were infected, and they made up the BSI group, whereas a second group included 36 patients who did not present BSI (non-BSI group). Patients from the first group presented BSI at median day 9 (25th and 75th quartiles were 5 and 11, respectively) after ICU admission. There were 60 pathogens that were isolated; 10 patients presented multi-bacterial bloodstream infection. A total of 77% (46 cases) of the isolates were gram-negative bacteria, and the remaining 23% (14 cases) were gram-positive (Table 4). Seventeen *A. baumannii* and ten *K. pneumoniae* isolates were PDR, and the rest were XDR, susceptible only to colistin, and colistin and aminoglycosides, respectively. The other isolates were MDR. Resistant *A. baumannii* and *K. pneumoniae* strains are endemic in our ICU, as previously described [13,14]. The high prevalence of resistance to antibiotics pathogens and the high antibiotic consumption may explain the abovementioned result. The mechanisms of resistance and transmission between patients were not studied.

### 3.1. Risk Factors for BSI

The baseline characteristics between groups are presented in Table 1. In Table 2 and Table 3, the characteristics of the patients before BSI or the total length of the ICU stay for the BSI group and non-BSI group are presented, respectively. Patients without dyslipidemia presented BSIs more frequently after univariate analysis (*p* < 0.05, Table 1).

### 3.2. Mortality and Morbidity Indices in Patients with BSI

Patients who presented BSI, when compared with patients who did not, had a longer length of ICU stay and a longer duration of mechanical ventilation and sedation (*p* < 0.05, Table 5). In this population, there was a trend towards increased mortality that did not reach statistical significance. Compared with non-survivors, survivors had lower baseline APACHE II and SOFA scores, lower D-dimers levels, and a higher baseline compliance of the respiratory system. They received anakinra less frequently and appropriate therapy more often (*p* < 0.05, Table 6). The multivariate analysis (Table 7) showed that the baseline APACHE II score [1.232 (1.017 to 1.493), *p* = 0.033] was the only independent risk factor for ICU mortality, while there was an indication towards increased mortality for patients who received anakinra [0.051 (0.003 to 1.026), *p* = 0.051].

## 4. Discussion

In the present study, we aimed to determine the incidence and to identify risk factors for BSI in critically ill, mechanically ventilated COVID-19 patients. Our results indicate that BSIs are frequent, since more than half of the patients were infected. Dyslipidemia occurs more often in non-infected patients. Furthermore, survivors had a significantly lower APACHE II score, and received anakinra less frequently when compared with non-survivors.

There are several studies on secondary infections in COVID-19 patients. Most of them include several types of infections, such as BSIs or infections of the respiratory tract. The populations included were usually mixed in terms of severity (hospitalizations both in ICUs and medical wards). Even in studies conducted in ICUs, patients may be under invasive mechanical ventilation or other forms of respiratory support, such as high flow oscillatory ventilation (HFOV) or non-invasive mechanical ventilation (NIV) [9,15,16,17,18]. To our knowledge, the present study is the first one to be conducted in the ICU, and all patients included were intubated and mechanically ventilated.

The incidence of BSIs in this study is higher when compared with our previously published data where patients did not present SARS-CoV-2 infection [13,14]. The results are also in accordance with other studies that report a higher number of COVID-19 patients with BSIs when compared with non-COVID-19 patients [9,19,20,21]. The profile of immune dysregulation and the higher percentage of COVID-19 patients that receive immunomodulatory agents may explain the finding.

The only risk factor for BSI that was identified in our study was dyslipidemia; more specifically, patients with dyslipidemia were protected from BSI. Certainly, this association does not imply a causative relationship. The concurrent administration of statins to these patients may play a role [22]. Data on this issue has not been reported previously in the literature; in this respect, this finding needs further investigation in the future with an appropriate methodology.

Despite the fact that the administration of antibiotics is widely known to be a factor responsible for infection, especially by multi-drug-resistant bacteria [13,14], we found no such evidence in this study. The shorter length of the hospital stay and the consequently lower use of antibiotics in comparison with non-COVID-19 patients, as well as the small number of participants in the present study, might be an explanation.

The results for the impact on secondary infections of immunosuppressive agents administered for the treatment of COVID-19 disease are inconclusive. There are studies where these agents are independently associated with increased nosocomial infections [9,17] and others that indicate no correlation [18]. Furthermore, there is no specific data for intubated and mechanically ventilated patients. In our study, the use of steroids, tocilizumab, or anakinra was not associated with BSIs. On the other hand, anakinra was associated with increased mortality. The etiology cannot be specified by the present study. Other factors related to this intervention, such as infections other than BSIs or different actions of anakinra, may be implicated.

BSIs did not affect mortality on a statistically significant level. The same result was identified by other studies [23]. The fact that the clinical outcome in severe COVID-19 patients is multifactorial may be an explanation for this. On the contrary, other indices of disease severity are affected: patients suffering from BSI had prolonged mechanical ventilation and a subsequent need for sedatives, and also a prolonged ICU length of stay, confirming the results from other studies [15,24,25]. Finally, the APACHE II score was higher in non-survivors. The relationship between the severity of illness and mortality is well established in several studies involving COVID-19 patients or non-COVID-19 patients [14,23].

The relationship between respiratory mechanics in patients with ARDS and mortality is not clear. According to the concept of patient self-inflicted lung injury (PSILI), the increased respiratory effort may generate lung injury in spontaneously breathing patients, leading to worse outcomes [26]. Consequently, early intubation and mechanical ventilation may prevent lung damage. Compliance of the respiratory (Crs) can be used as an indicator of the lung injury in ARDS patients. Higher values of Crs indicate less lung injury. In our study, survivors presented higher Crs after intubation and during ICU admission, but Crs was not an independent factor for mortality after the multivariate analysis. The fact that a higher APACHE II score predicted a worse outcome when compared to Crs alone suggests that the overall severity of multi-organ failure is more important than isolated respiratory mechanics.

This study presents limitations. It was performed at a single center, and the results should therefore be interpreted cautiously. The number of participants was relatively small. The fact that most of our pathogens are pan-drug-resistant, as previously described [13,14], may limit the generalizability of the results. However, the findings of this study may form the basis for a further investigation in the future.

## 5. Conclusions

A considerable percentage of intubated and mechanically ventilated patients with SARS-CoV-2 infection present BSI. Fever or reduced serum concentrations of inflammatory markers may make the diagnosis of BSI difficult if immunomodulatory drugs are used; therefore, close monitoring may improve the outcome. Finally, further studies are required to confirm the aforementioned findings.

## 6. Definitions

BSI was defined according to Center of Disease Control (CDC) criteria [27]. Previous hospitalization was defined as the admission to hospitals or other healthcare facilities for >48 h during the last three months. Antibiotics against Gram (+) bacteria include teicoplanin, daptomycin, vancomycin, and linezolid. As appropriate therapy was considered to be the administration of in vitro active antibiotics for at least 48 h. EUCAST breakpoints were used for susceptibility testing. SARS-CoV-2 infection was confirmed by reverse transcription polymerase chain reaction (PCR) with nasopharyngeal swabs. No genetic testing was performed. Patients’ treatment decision was at the attending physician’s discretion, and thus antibiotic combinations were different among patients. Only a single dose of tocilizumab was administered. No antibiotics were given as a prophylaxis in the ICU. Pandrug-resistant (PDR) was defined as a pathogen that was nonsusceptible to all agents in all antimicrobial categories, extensively drug-resistant (XDR) as a pathogen that was susceptible to only one or two antimicrobial categories, and finally multidrug-resistant (MDR) as a pathogen that was resistant to at least one agent in three or more drug classes.

## Figures and Tables

**Table 1 antibiotics-11-01053-t001:** Baseline characteristics during ICU admission.

	BSI Group (N = 48)	Non-BSI Group (N = 36)	*p*
Sex (Male)	32 (67)	24 (67)	-
Age (years)	69 (57, 76)	71 (66, 76)	0.274
APACHE II score	13 (10, 19)	16 (12, 22)	0.070
SOFA score	7 (7, 8)	8 (7, 10)	0.204
PaO2/FiO2 ratio	161 (120, 198)	154 (118, 199)	0.953
Crs	36 (30, 44)	34 (22, 45)	0.472
D-dimers	767 (522, 1134)	1042 (566, 2156)	0.093
Lymphocyte Count	640 (400, 847)	600 (413, 790)	0.772
Ferritin	976 (510, 1707)	1488 (861, 2455)	0.068
Hospitalization in the last 3 months	1 (2)	1 (3)	-
Days of hospitalization before ICU admission	1 (1, 4)	2 (1, 7)	0.084
Diabetes Mellitus	18 (38)	10 (28)	0.483
Chronic Lung disease	11 (23)	11 (31)	0.483
Chronic Heart Failure	5 (11)	3 (8)	-
Chronic Renal failure	2 (4)	3 (8)	0.647
Neurological disease	6 (13)	6 (17)	0.754
Arterial Hypertension	26 (54)	26 (72)	0.114
Malignancy	2 (4)	3 (8)	0.647
Dyslipidemia	10 (21)	16 (44)	0.031
Coronary Heart Disease	11	7 (19)	0.792
Autoimmune	2	2 (6)	-

Data is presented as median (25%, 75% quartiles) or n (%); BSI, Blood Stream Infection; ICU, Intensive Care Unit; APACHE, Acute Physiology and Chronic Health Evaluation; SOFA, Sequential Organ Failure Assessment; Crs, Compliance of the respiratory system; *p*, comparison between the two groups. Results by univariate analysis, chi square test, or Fisher’s exact test for categorical variables and by Mann–Whitney *U* test for continuous variables.

**Table 2 antibiotics-11-01053-t002:** Clinical characteristics in the ICU before BSI.

	BSI Group (N = 48)	Non-BSI Group (N = 36)	*p*
MV duration (days)	8 (4, 11)	7 (4, 12)	0.895
Invasive procedures	2 (4)	2 (6)	-
CVVHDF use	9 (19)	8 (22)	0.786
CVVHDF duration (days)	5 (2, 9)	4 (3, 5)	0.843
Steroids	40 (83)	33 (92)	0.338
Tocilizumab	11 (23)	13 (36)	0.226
Anakinra	7 (15)	12 (33)	0.064
Remdesivir	12 (25)	10 (28)	0.806

Data is presented as median (25%, 75% quartiles) or n (%); BSI, Blood Stream Infection; ICU, Intensive Care Unit; MV, mechanical ventilation; CVVHDF, Continuous veno-venous hemodiafiltration; Invasive procedures, gastroscopy, colonoscopy, or bronchoscopy; *p*, comparison between the two groups. Results by univariate analysis, chi square test, or Fisher’s exact test for categorical variables and by Mann–Whitney *U* test for continuous variables.

**Table 3 antibiotics-11-01053-t003:** Antibiotics administered to participants before BSI.

	BSI Group (N = 48)	Non-BSI Group (N = 36)	*p*
Antibiotics during the last 3 months	1 (2)	0 (0)	-
Antibiotics during hospitalization prior to infection	48 (100)	36 (100)	-
Use of Carbapenems	22 (45)	19 (53)	0.660
Use of Antipseudomonal Penicillins	13 (27)	9 (25)	0.155
Use of Quinolones	24 (50)	18 (50)	-
Use of Cephalosporins 3d generation	23 (48)	12 (33)	0.263
Use of Ceftarolin	20 (42)	12 (33)	0.500
Use of Colistin	25 (52)	16 (44)	0.516
Use of Tygecycline	21 (44)	19 (52)	0.509
Use of Aminoglycosides	7 (15)	3 (8)	0.504
Gram (+) antibiotics	35 (73)	29 (80)	0.446
TMP/SMX	10 (20)	15 (42)	0.551
Use of CAZ-AVI	9 (19)	2 (6)	0.105

Data is presented as median (25%, 75% quartiles) or n (%); BSI, Blood Stream infection; TMP/SMX, trimethoprim-sulfamethoxazole; CAZ-AVI, ceftazidime-avibactam; Gram (+) antibiotics, teicoplanin, daptomycin, vancomycin, and linezolid; *p*, comparison between the two groups. Results by univariate analysis, chi square test, or Fisher’s exact test for categorical variables and by Mann–Whitney *U* test for continuous variables.

**Table 4 antibiotics-11-01053-t004:** Pathogens detected in blood stream infections.

Pathogen	Number of Cases (N = 60)
*Acinetobacter baumannii*	20 (33%)
*Klebsiella pneumonia*	19 (32%)
*Stenotrophomonas maltophilia*	3 (5%)
*Pseudomonas aeruginosa*	2 (3%)
*Proteus mirabilis*	1 (2%)
*Serratia marcescens*	1 (2%)
*Enterococcus* spp.	14 (23%)

**Table 5 antibiotics-11-01053-t005:** Outcomes.

	BSI Group (N = 48)	Non-BSI Group (N = 36)	*p*
ICU length of stay (days)	18 (14, 26)	7 (5, 12)	<0.000
Death	35 (73)	20 (56)	0.111
MV duration (days)	14 (18, 23)	7 (4, 12)	<0.000
Duration of Sedation (days)	13 (9, 18)	7 (4, 11)	<0.000

Data is presented as median (25%, 75% quartiles) or n (%); BSI, Blood Stream Infection; ICU, intensive care unit; MV, mechanical ventilation; *p*, comparison between the two groups. Results by univariate analysis, chi square test, or Fisher’s exact test for categorical variables and by Mann–Whitney *U* test for continuous variables.

**Table 6 antibiotics-11-01053-t006:** Characteristics of survivors and non-survivors in the ICU.

	Survivors (N = 29)	Non-Survivors (N = 55)	*p*
Sex (Male)	18 (62)	38 (69)	0.627
Age (years)	70 (61, 74)	70 (62, 77)	0.296
APACHE II score	12 (11, 15)	17 (12, 22)	0.001
SOFA score	7 (6, 8)	8 (7, 9)	0.009
BSI	13 (45)	35 (64)	0.111
Diabetes Mellitus	6 (20)	22 (40)	0.091
Chronic Lung disease	6 (20)	16 (29)	0.447
Chronic Heart Failure	0 (0)	8 (15)	0.046
Chronic renal disease	1 (3)	4 (7)	0.665
Neurological disease	3 (10)	9 (16)	0.531
Arterial Hypertension	16 (55)	36 (65)	0.479
Malignancy	3 (10)	2 (4)	0.335
Autoimmune disease	1 (3)	3 (5)	-
Dyslipidemia	10 (34)	16 (29)	0.628
Coronary Artery Disease	4 (14)	14 (25)	0.271
Total ICU length of stay (days)	14 (8, 25)	13 (7, 18)	0.349
Total MV duration (days)	11 (6, 19)	7 (4, 12)	0.349
Total duration of sedation (days)	9 (5, 15)	7 (4, 12)	0.193
Appropriate Therapy	9/13 (69)	12/35 (34)	0.049
Steroids	24 (83)	49 (89)	0.501
Tocilizumab	7 (24)	17 (31)	0.615
Anakinra	2 (7)	17 (31)	0.014
Remdesivir	9 (31)	13 (24)	0.602
PaO2/FiO2 ratio	164 (137, 204)	125 (99, 156)	0.078
Crs	40 (30, 47)	34 (27, 42)	0.032
D-dimers	659 (473, 1023)	1180 (1140, 4578)	0.027
Lymphocyte Count	500 (390, 845)	600 (435, 738)	0.556
Ferritin	941 (554, 1997)	1617 (1058, 2795)	0.204

Data is presented as median (25%, 75% quartiles) or n (%); BSI, Blood Stream Infection; ICU, intensive care unit; APACHE, Acute Physiology and Chronic Health Evaluation; SOFA, Sequential Organ Failure Assessment; MV, mechanical ventilation; Crs, compliance of the respiratory system; *p*, comparison between the two groups. Results by univariate analysis, chi square test, or Fisher’s exact test for categorical variables and by Mann–Whitney *U* test for continuous variables.

**Table 7 antibiotics-11-01053-t007:** Multivariate analysis.

	Odds Ratio	95% CI	*p*
APACHE II Score	1.232	1.017–1.493	0.033
SOFA score	0.460	0.203–1.044	0.063
Dyslipidemia	0.000	0.000	0.999
Appropriate therapy	4.553	0.855–24.257	0.076
Anakinra	0.051	0.003–1.026	0.051
Compliance of respiratory system	0.983	0.917–1.053	0.628
D-Dimers	1.000	1.000–1.001	0.635

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
