# Peer review of "Incidence and Risk Factors for Blood Stream Infection in Mechanically Ventilated COVID-19 Patients"

_antibiotics, 2022, doi:10.3390/antibiotics11081053_

Round 1

Reviewer 1 Report

The authors have done a great job by determining the incidence and risk factors of blood infection sin COVID-19 patients who are on ventilation. There are some flaws that shall be addressed .some of them are:
1.In title of the manuscript individuals shall be replaced by patients.

2. In abstract line 2 the word BSi shall be written complete and not in abbreviation.

3. Line 14 in the abstract section needs complete rephrasing as it is not proper understandable.

4. Outcome must be written clearly.

5. Sections of definition may please be replaced by letters of abbreviations and shall be shifted at the end of the manuscript.

6. Add some recommendations at the end.

7. Yet I am not a native speaker of English language but still I recommend that the English language needs touching up in a major way. The article needs to be rewritten in readable English. Many sentences are confusing, do not lead to scientific meaning, and can be found starting in lower case, and upper case can be detected in the middle of sentences without proper nouns.

Author Response

Dear Reviewer,

 We were pleased that you are interested in a revised version of our manuscript ‘Incidence and Risk factors for Blood Stream Infection in Mechanically Ventilated COVID-19 Individuals’.

We wish to thank you for your valuable comments that helped us to improve our manuscript.

 In more detail:

COMMENTS TO THE AUTHOR (IN CAPITALS)

Response of the authors (lowercase in blue)

 1.IN TITLE OF THE MANUSCRIPT INDIVIDUALS SHALL BE REPLACED BY PATIENTS.

In title the word ‘Individuals’ is replaced by ‘Patients’

  1. IN ABSTRACT LINE 2 THE WORD BSI SHALL BE WRITTEN COMPLETE AND NOT IN ABBREVIATION.

BSI is written complete.

  1. LINE 14 IN THE ABSTRACT SECTION NEEDS COMPLETE REPHRASING AS IT IS NOT PROPER UNDERSTANDABLE.

Line 14 is rephrased. Please refer to lines 13-14.

  1. OUTCOME MUST BE WRITTEN CLEARLY.

The incidence of BSIs which is the primary outcome of the study is written clearly in the Abstract. Please refer to line 17.

  1. SECTIONS OF DEFINITION MAY PLEASE BE REPLACED BY LETTERS OF ABBREVIATIONS AND SHALL BE SHIFTED AT THE END OF THE MANUSCRIPT.

Definition is shifted at the end of the manuscript according to your advise.

  1. ADD SOME RECOMMENDATIONS AT THE END.

At the end of the manuscript recommendations about patients’ monitoring and the need for further studies were added. Please refer to lines 198-201.

  1. YET I AM NOT A NATIVE SPEAKER OF ENGLISH LANGUAGE BUT STILL I RECOMMEND THAT THE ENGLISH LANGUAGE NEEDS TOUCHING UP IN A MAJOR WAY. THE ARTICLE NEEDS TO BE REWRITTEN IN READABLE ENGLISH. MANY SENTENCES ARE CONFUSING, DO NOT LEAD TO SCIENTIFIC MEANING, AND CAN BE FOUND STARTING IN LOWER CASE, AND UPPER CASE CAN BE DETECTED IN THE MIDDLE OF SENTENCES WITHOUT PROPER NOUNS.

English editing of the manuscript was performed by a professional English Literature graduate.

We hope that you will find our answers thorough and comprehensive.

Thank you in advance

Reviewer 2 Report

1. The authors should be consistent with the use of "SARS-CoV-2" versus "SARS-CoV-19". The latter is uncommon and potentially inaccurate. 

2. Given the current status of COVID-19, readers may find it helpful to know which variants were predominant or included in the study. Also consider providing an exact date range.  

3. In the results section, consider organizing the infecting pathogens into a table. At present, it is difficult to follow in paragraph form. Moreover, how many patients had what would typically be deemed a contaminant (eg coagulase-negative Staphylococcus)?  

4. In the results section, drug resistance (eg XDR) is defined. Consider adding this to the definitions section. 

5. In Table 2, more detail regarding COVID interventions may be useful to readers. The authors highlight the use of anakinra; however, how many patients received multiple doses of tocilizumab? Did patients receive tocilizumab with active infection? If so, how many? 

6. In Table 2, the CVVHDF patients had a median of 0 days? Please confirm this is accurate. 

7. In Table 3, Gram positive antibiotics as a group should be teased out. The authors suggest Enterococcus was the most common finding, thus was vancomycin predominantly used? Did it have an effect on the findings? 

8. In Table 3, if a patient received a dose were they counted? This data needs more detail to be meaningful. If a patient received a single dose of a FQ versus 7 days of cefepime I would not consider them the same with respect to impact on resistance. 

9. The authors appear to allude to dyslipidemia having a relationship with survival and subsequently use of statins. Current evidence with use of statins in COVID-19 is "murky" at best. The authors should strongly consider tempering the language in the conclusion. 

Author Response

Dear Reviewer,

 We were pleased that you are interested in a revised version of our manuscript ‘Incidence and Risk factors for Blood Stream Infection in Mechanically Ventilated COVID-19 Individuals’.

We wish to thank you for your valuable comments that helped us to improve our manuscript.

 In more detail:

COMMENTS TO THE AUTHOR (IN CAPITALS)

Response of the authors (lowercase in blue)

 THE AUTHORS SHOULD BE CONSISTENT WITH THE USE OF "SARS-COV-2" VERSUS "SARS-COV-19". THE LATTER IS UNCOMMON AND POTENTIALLY INACCURATE.

You are right. ‘SARS-CoV-19’ is corrected to ‘SARS-CoV-2’ throughout the text.

  1. GIVEN THE CURRENT STATUS OF COVID-19, READERS MAY FIND IT HELPFUL TO KNOW WHICH VARIANTS WERE PREDOMINANT OR INCLUDED IN THE STUDY. ALSO CONSIDER PROVIDING AN EXACT DATE RANGE.

Thank you for your comment. Unfortunately, there was not the availability for genetic testing. A comment was added in the text. Please refer to line 209. The exact date range was added (line 54)

  1. IN THE RESULTS SECTION, CONSIDER ORGANIZING THE INFECTING PATHOGENS INTO A TABLE. AT PRESENT, IT IS DIFFICULT TO FOLLOW IN PARAGRAPH FORM. MOREOVER, HOW MANY PATIENTS HAD WHAT WOULD TYPICALLY BE DEEMED A CONTAMINANT (EG COAGULASE-NEGATIVE STAPHYLOCOCCUS)?

Thank you for your advise. A new table was added with the isolated pathogens, and also they were removed from the results section (table 4). CoNS were isolated in five cases that were excluded because they were considered as contamination.

  1. IN THE RESULTS SECTION, DRUG RESISTANCE (EG XDR) IS DEFINED. CONSIDER ADDING THIS TO THE DEFINITIONS SECTION.

All definitions about antibiotic resistance were removed to ‘Definitions’ (lines 212-216)

  1. IN TABLE 2, MORE DETAIL REGARDING COVID INTERVENTIONS MAY BE USEFUL TO READERS. THE AUTHORS HIGHLIGHT THE USE OF ANAKINRA; HOWEVER, HOW MANY PATIENTS RECEIVED MULTIPLE DOSES OF TOCILIZUMAB? DID PATIENTS RECEIVE TOCILIZUMAB WITH ACTIVE INFECTION? IF SO, HOW MANY? 

Thank you for your remark. Based on hospital local protocol patients received a single dose of tocilizumab early during hospitalization. The presence of secondary infection was contraindication to tocilizumab administration. A relevant comment was added (line 211).

  1. IN TABLE 2, THE CVVHDF PATIENTS HAD A MEDIAN OF 0 DAYS? PLEASE CONFIRM THIS IS ACCURATE.

Yes, median days of CVVHDF was zero. This was because many patients did not receive CVVHDF and the distribution was skewed. In light of your comment in order to avoid confusion we now present CVVDHDF duration only in those who received renal replacement therapy.

  1. IN TABLE 3, GRAM POSITIVE ANTIBIOTICS AS A GROUP SHOULD BE TEASED OUT. THE AUTHORS SUGGEST ENTEROCOCCUS WAS THE MOST COMMON FINDING, THUS WAS VANCOMYCIN PREDOMINANTLY USED? DID IT HAVE AN EFFECT ON THE FINDINGS?

Thank you for your notice. In table 3 are listed the antibiotics that were prescribed to the patients before BSI. Among them daptomycin was used more frequently. Vancomycin was used in totally 14 patients. Neither daptomycin nor vancomycin nor other antibiotic was associated significantly with BSI or other outcome in analysis.

  1. IN TABLE 3, IF A PATIENT RECEIVED A DOSE WERE THEY COUNTED? THIS DATA NEEDS MORE DETAIL TO BE MEANINGFUL. IF A PATIENT RECEIVED A SINGLE DOSE OF A FQ VERSUS 7 DAYS OF CEFEPIME I WOULD NOT CONSIDER THEM THE SAME WITH RESPECT TO IMPACT ON RESISTANCE.

You are right that tis point may add confusion. Data are available to be added. However, in light of the suggestion of Reviewer 1, days of administration of each antibiotic were removed because there was no statistical difference between the two groups.

  1. THE AUTHORS APPEAR TO ALLUDE TO DYSLIPIDEMIA HAVING A RELATIONSHIP WITH SURVIVAL AND SUBSEQUENTLY USE OF STATINS. CURRENT EVIDENCE WITH USE OF STATINS IN COVID-19 IS "MURKY" AT BEST. THE AUTHORS SHOULD STRONGLY CONSIDER TEMPERING THE LANGUAGE IN THE CONCLUSION.

You are right. It is not proven that dyslipidemia and the subsequent use of statins reduces the rate of BSIs or improves survival. Thus, we removed dyslipidemia from ‘Conclusion’.

 We hope that you will find our answers thorough and comprehensive.

Round 2

Reviewer 2 Report

1. How is "appropriate therapy" defined? This is relevant especially given your date range. 

2. In Table 7, you do not define ExpB. Please clarify this. In this analysis, should heart failure be included given the findings?  

3. The authors should consider commenting on the level of drug-resistance in isolated organisms.  

Author Response

This manuscript is a resubmission of an earlier submission. The following is a list of the peer review reports and author responses from that submission.

Round 1

Reviewer 1 Report

The research article seeks to determine the risk factors of mechanically ventilated COVID-19 patients for developing blood stream infections. the authors failed to identify any clear rick factors that could have lead to BSIs. I have several concerns with this rather brief report. First, the authors fail to discuss if there are any bacterial species that are endemic to the ICU. Considering the high number of BSIs, it is possible that some of the common pathogens (Klebsiella and Acinetobacter) in the study were spreading among patients close together. 

They do not disclose why these patients were placed on different antibiotic prophylaxes. Or, whether some of the patients were being treated with different combinations of antibiotics. 

The authors do not discuss the resistance nature of these bacterial species despite stating that they have tested them in the methods section.

In the discussion they mention that there was a clear statistical trend between BSI and mortality (p of 0.051). However, in table 5 the p value is 0.111.

Reviewer 2 Report

The study is good but the authors improve the following things;

  1. Introduction is short.
  2. Improve the sample size and
  3. Improve the discussion section by adding some more references

Reviewer 3 Report

Dear authors,

The presented work shows interest in demonstrating a comparison to investigate the risk factors for the first episode of bloodstream infection in mechanically ventilated patients with COVID-19. However, the conclusions and achievements made do not show conclusive or relevant results regarding future treatments, which are handled today. The study must be strengthened with analyzes that include other hospital centers, to demonstrate that the results improve their statistical power.

Reviewer 4 Report

Manuscript - Risk factors for the first episode of Blood Stream Infection in Mechanically Ventilated COVID-19 Patients

Title: The authors should avoid using the word “patients”. Considered the following title: “Risk factors for the first episode of Blood Stream Infection in Mechanically Ventilated Coronavirus Disease (COVID)-19 individuals

Also, the word “patients” should be changed to “individuals”

To delete the word “Abstract”

Abstract

The authors should redo the text, and it was not necessary to divide it into topics.

To change “d-dimers” to “D-dimers”

In the abstract, the authors should focus on what is new, and in the main objective, in that sense, it is crucial to point out the importance of BSI in individuals on mechanical ventilation due to COVID-19. Also, it is important to present the number of deaths in both scenarios (with and without BSI). To include the microbiology profile. Finally, the conclusion “Severity of illness affects mortality” should be deleted because it is redundant.

Keywords

To include other words, such as SARS-CoV-2

Introduction:

To correct the word “(SARSCoV-2)” to “(SARS-CoV-2)”

To delete the first paragraph

To include more information about the BSI in individuals with COVID-19 (e.g., epidemiology, treatment, duration, clinical importance, disease evolution, microbiology, and others).

Methods

To change “COVID-19 infection” to “SARS-CoV-2 infection”

The following sentence, “For survivors and non survivors several characteristics that might affect mortality were recorded: age, sex, characteristics of the respiratory system, severity of illness, comorbidities, invasive procedures, total duration of mechanical ventilation (MV) and sedation, and BSI.” repeated some markers. I believe that the authors should only cite the exclusive ones.

Results

Did the authors collect all the data for all individuals? Is there any missing data?

Table 1 and Table 2. It was used bivariate analyses – BSI groups vs. clinical features. Also, the authors could write the names of the statistical tests.

Table 2. To revise the legend. Some abbreviations were used only in table 1.

Table 3. The authors can remove the days for each antibiotic. This information did not have great importance for the study.

Topic “Mortality and Morbidity Indices in Patients with ABRC Infection”. What is the meaning of ABRC?

Importantly, revise all Table legends.

The multivariate should be presented as a table. Also, the authors should describe it better in the methods and results sections.

To include a sample power calculation. Also, all the markers used in the results should be cited in the methods section.

Finally, the authors should focus on the major study questions – BSI in ICU individuals due to COVID-19

Discussion

To delete the results. For example: “BSIs did not affect mortality to a statistically significant level although there was a clear trend (p=0.051) towards higher mortality for the BSI group. The same result was identified by other studies (17). The fact that the clinical outcome in severe COVID-19 patients is multifactorial may be an explanation.”

Also, it is important to discuss all findings based on literature and to demonstrate the possible generalizability of the results.

Conclusion

The conclusion should be revised to corroborate with the primary study question.